# LAMBDABEAM: Neural Program Search with Higher-Order Functions and Lambdas

**Kensen Shi**
Google DeepMind
kshi@google.com

**Hanjun Dai**
Google DeepMind
hadai@google.com

**Wen-Ding Li**
Cornell University
wl678@cornell.edu

**Kevin Ellis**
Cornell University
kellis@cornell.edu

**Charles Sutton**
Google DeepMind
charlessutton@google.com

## Abstract

Search is an important technique in program synthesis that allows for adaptive strategies such as focusing on particular search directions based on execution results. Several prior works have demonstrated that neural models are effective at guiding program synthesis searches. However, a common drawback of those approaches is the inability to handle iterative loops, higher-order functions, or lambda functions, thus limiting prior neural searches from synthesizing longer and more general programs. We address this gap by designing a search algorithm called LAMBDABEAM that can construct arbitrary lambda functions that compose operations within a given DSL. We create semantic vector representations of the execution behavior of the lambda functions and train a neural policy network to choose which lambdas to construct during search, and pass them as arguments to higher-order functions to perform looping computations. Our experiments show that LAMBDABEAM outperforms neural, symbolic, and LLM-based techniques in an integer list manipulation domain.

## 1 Introduction

Program synthesis involves finding a program meeting a given specification of what the program should do [21, 18]. When the specification is in the form of input/output examples, known as programming by example (PBE), combinatorial search has been an especially popular technique [2, 30, 3, 32, 4, 27, 33]. Learning can also play a key role in PBE, because well-designed search algorithms can learn to adapt to information collected during the ongoing search, such as execution results or other analyses of candidate programs considered so far. This information can be used to prune redundant parts of the search space or focus on parts deemed more promising. For example, DeepCoder [3] and TF-Coder [32] use neural models to define a search space that is explored by traditional non-neural search, while BUSTLE [27], CROSSBEAM [33], and Execution-Guided Synthesis [7] use neural models to guide the search process itself. However, those prior works are unable to generate programs with arbitrary looping computations, whether implemented via loop control structures or through the use of higher-order functions with arbitrary lambda functions.[1] Even though large language models (and other sequence models) can output programs with loops and are very effective at synthesizing programs from natural language [6], PBE demands a more systematic search strategy that adapts to valuable information like execution results during the search.

---

[1]DeepCoder [3] supports higher-order functions but only a small set of hardcoded lambda functions. Execution-Guided Synthesis [7] supports variable-free while loops, but not loops with an iteration variable.

37th Conference on Neural Information Processing Systems (NeurIPS 2023).

The fundamental question explored in this paper is whether a neural program synthesis search policy can learn to reason about lambdas and higher-order functions, which would enable the synthesis of arbitrary looping computations that were not previously possible with neural synthesis search techniques that rely on intermediate expression evaluation. Previous work [27, 33] has shown that neural models can effectively guide search when every candidate program can be evaluated to produce a concrete value for the model to inspect, allowing it to make decisions based on comparisons between explored values and the desired output. Lambda functions however are extremely different: they represent plans of functionality to be performed later, without specifying the context in which this functionality will be used. As a result, reasoning about lambdas requires a more abstract form of planning. Without knowing how the lambda might be used later, the search policy must understand the different behaviors of lambdas, predict whether a lambda will be useful for a given task to prioritize search directions, and recognize when and how to use lambdas within higher-order functions to actually perform useful computations.

In order to design a neural search algorithm that handles lambdas and higher-order functions, we address some key difficulties. One challenge is in the algebraic representation and subsequent manipulation of lambdas. We want to represent "practically equivalent" lambdas like $\lambda x.\ x + 1$, $\lambda y.\ y + 1$, and $\lambda x, y.\ x + 1$ in a canonical way to prune the search space. If $\lambda x.\ x + 1$ is their canonical representation, then how can we reuse that lambda to create a new lambda such as $\lambda x, y.\ (x + 1) \times (y + 1)$ where the "$\circ + 1$" functionality is used in different ways? We address this challenge by defining a new MERGE operation that combines lambda expressions into larger ones while allowing for variable renaming and adhering to other representational constraints, therefore enabling a bottom-up search algorithm to systematically build larger lambdas from existing ones.

A second difficulty is in the encoding of lambdas when used as inputs to neural models. A naive representation would be to encode the code tokens, but slight changes in the code could lead to drastic differences in the lambda's behavior. Instead, we introduce a method of encoding lambdas that more directly reflects their execution semantics. We do this using *property signatures* [26] in a way agnostic to how the lambda is used later (i.e., what inputs are given to the lambda by a higher-order function), but still analyzing the lambda's behavior in the context of the current PBE task. One conclusion of our work is that this encoding does enable neural models to reason about lambdas effectively.

We present a new neural synthesis search method called LAMBDABEAM, which combines our solutions to these challenges within the search framework of the recent work CROSSBEAM [33]. CROSSBEAM performs a bottom-up search applying DSL operations to previously-explored values, using a neural search policy to choose the operation's arguments with a pointer network. Thus, in LAMBDABEAM, the neural policy is able to reason about lambdas by choosing which ones to construct and when to use them, such that the search direction is tailored to the synthesis task.

We demonstrate the effectiveness of LAMBDABEAM in the DeepCoder [3] domain of integer list manipulation. We extend the DeepCoder DSL by adding many first-order operations, keeping its higher-order functions, and replacing its limited set of hardcoded lambda functions with arbitrary lambdas using compositions of other DSL operations. Using a benchmark suite containing 100 natural hand-crafted evaluation tasks and 100 synthetically-generated tasks, we experimentally show that LAMBDABEAM outperforms prior approaches including state-of-the-art symbolic search, neural sequence models trained from scratch, and a 62 billion parameter large language model (LLM). We release our LAMBDABEAM code and trained model checkpoints at `https://github.com/ellisk42/LambdaBeam`.

## 2    Background

**Programming By Example**    *Programming by Example* (PBE) is the task of synthesizing programs that satisfy a given set of input/output (I/O) examples. In this task, we have a domain-specific language (DSL) $\mathcal{L}$ describing a space of programs, and a set of example inputs $\mathcal{I} = \{I_1, \ldots, I_N\}$ and corresponding outputs $\mathcal{O} = \{O_1, \ldots, O_N\}$. The goal is to find a program $P \in \mathcal{L}$ such that $P(I_i) = O_i$ for all $i \in \{1, \ldots, N\}$. The DSL $\mathcal{L}$ describes atomic values (constants and input variables) and operations that can be applied to arguments to produce new values. Programs in $\mathcal{L}$ are arbitrarily-nested compositions of operations applied to atomic values or other such compositions.

**$\lambda$-Calculus**    The lambda calculus [10, 28] is a formalism for universal computation. A lambda calculus *term* is either a variable, function application, or a lambda abstraction. Lambda abstractions

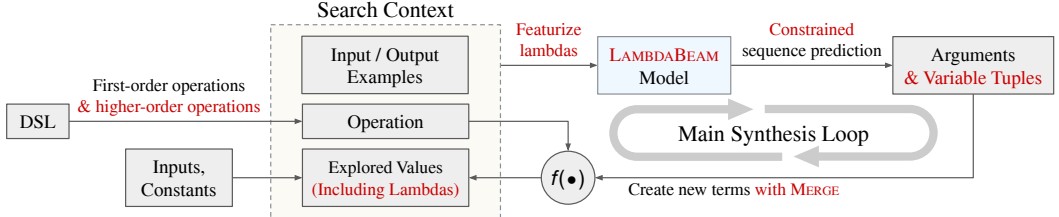

Figure 1: Overview of LAMBDABEAM which builds upon the prior work CROSSBEAM (new elements in LAMBDABEAM shown in red).

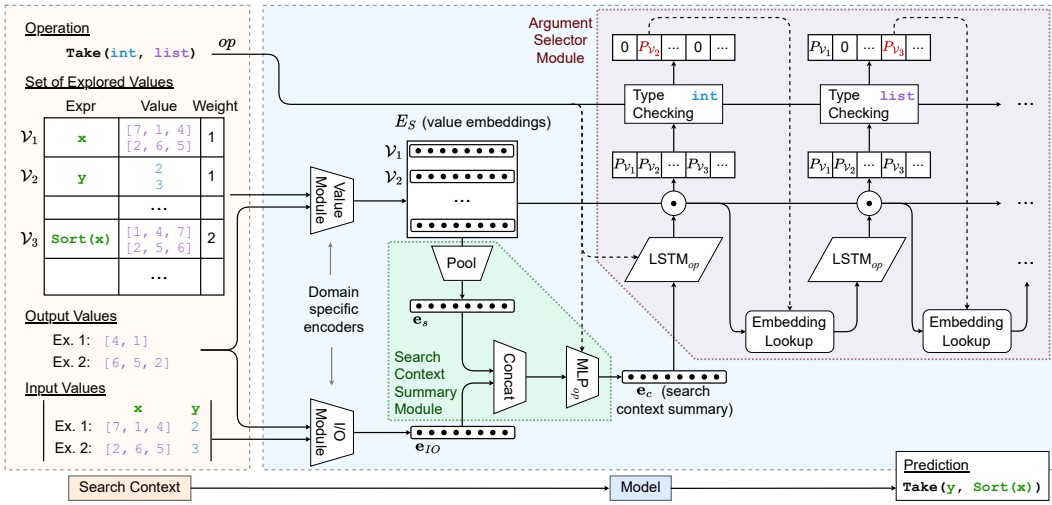

Figure 2: Diagram of the LAMBDABEAM model architecture, closely mirroring that of CROSSBEAM.

define new functions by introducing new lexically-scoped variables, as well as a function body (which is itself a term). We consider lambda abstractions that are allowed to introduce multiple variables at once (we do not "Curry" our lambdas). Terms in the lambda calculus are constructed recursively from smaller terms, resulting in tree-like structures like in Figure 3(a). We extend $\lambda$-calculus with domain-specific primitives (add, sort, map, filter, etc.) as well as a basic type system known as simply-typed lambda calculus; see [28] for details.

**Property Signatures** During a neural program synthesis search, many different expressions are encountered, possibly including lambda functions with arbitrary input and output types. How might a neural model learn to reason about these expressions? One option is to represent expressions as source code and apply standard text encoders like recurrent neural networks or Transformers. However, two expressions with similar source code might have very different execution behavior, or two syntactically distinct expressions might have identical semantics. An alternative approach that is more semantically aware is based on *property signatures* [26, 27], inspired by property-based testing in the programming languages literature [16]. Unary properties describe a single value by mapping it to a boolean, such as whether a list is sorted, or whether a string is empty. Binary properties can describe the relationship between the input and output of a PBE task, such as whether the input and output lists have the same length. In either case, given a list of $k$ property functions, we can evaluate all property functions to form a vector of length $k$ called a *property signature*. Each element of the signature is either True, False, or N/A.[2] This vector may be used as input to a deep neural network. Furthermore, given multiple executions of an expression (e.g., over different examples), we can identify how often each property holds, leading to a more granular representation.

**CROSSBEAM** Our work LAMBDABEAM builds upon the prior work CROSSBEAM [33]. Both systems have a similar overall structure, illustrated in Figure 1 with differences shown in red. The core

---

[2]A property might be *not applicable* (N/A) if it does not apply to the types of objects currently under consideration, or if the property is inconclusive because the code execution resulted in an error.

| (a) Term as a tree | (b) Term as straightline code | (c) Built using MERGE |
|---|---|---|

$\lambda v_1$
|
map

$\lambda u$     sort

add     $x$

$v_1$   square
|
$u$

$\mathtt{t}_1 \leftarrow x$

$\mathtt{t}_2 \leftarrow \lambda v_1.\ \mathrm{square}(v_1)$

$\mathtt{t}_3 \leftarrow \lambda v_1, v_2.\ \mathrm{add}(v_1, \mathtt{t}_2(v_2))$

$\mathtt{t}_4 \leftarrow \mathrm{sort}(\mathtt{t}_1)$

$\mathtt{t}_5 \leftarrow \lambda v_1.\ \mathrm{map}(\lambda u.\ \mathtt{t}_3(v_1, u), \mathtt{t}_4)$

atomic term from DSL

$\mathrm{MERGE}(\mathrm{square}, v_1, [\,])$

$\mathrm{MERGE}(\mathrm{add}, v_1, [\,], \mathtt{t}_2, [v_2])$

$\mathrm{MERGE}(\mathrm{sort}, \mathtt{t}_1, [\,])$

$\mathrm{MERGE}(\mathrm{map}, \mathtt{t}_3, [v_1, u], \mathtt{t}_4, [\,])$

Figure 3: Different ways of representing the lambda expression $\lambda v_1.\ \mathrm{map}(\lambda u.\ v_1 + u^2, \mathrm{sort}(x))$, where $x$ is an input variable. The model predicts blue term pointers and variables, the search tries every red operation, and unshaded code tokens in (b) are known from the other shaded tokens.

idea in CROSSBEAM is to use a neural policy network to guide a bottom-up search over programs, where execution results of explored expressions provide a powerful signal for the policy to guide the search by exploring new expressions whose values are closer to the intended output.

In CROSSBEAM, a table stores the expressions explored so far in search, along with their execution values when run on the I/O examples. As diagrammed in Figure 2, the Value Module encodes each explored value into a vector that is stored in a matrix $E_S$. Meanwhile, the I/O Module encodes the I/O examples into an analogous vector $\mathbf{e}_{IO}$. Then, the Search Context Summary Module combines the value encodings $E_S$ and the I/O encoding $\mathbf{e}_{IO}$ to produce a latent "summary" $\mathbf{e}_c$ of the search context so far. From this, the Argument Selector Module, which is a recurrent pointer network [35], predicts an argument list for a DSL operation via a sequence of pointers into the table of explored values, thus generating a new expression to explore. By structuring the search in this way, the neural policy is able to take a "hands on" role during search, using information in the search context to choose which programs should be explored next.

The network is trained on-policy using imitation learning. Each item in the training set consists of a set of I/O examples and a target program. During training, we run the search algorithm on the I/O examples where at each step the policy network proposes values to explore by predicting argument lists for DSL operations. We then apply a softmax loss that encourages the policy network to make predictions that would lead to progress according to the target program, instead of other argument lists that the policy network actually proposed.

During evaluation, getting argument lists via beam search as in training can lead to the search stalling if all new values are semantically equivalent to values already seen, since beam search is deterministic if the value set is unchanged. To address this, during evaluation CROSSBEAM randomly samples different argument lists using UniqueRandomizer [31] to avoid duplicate samples.

## 3 The LAMBDABEAM Approach

### 3.1 Building λ-Terms

LAMBDABEAM constructs terms in the lambda calculus during its search. A natural design choice is to construct terms in a way that avoids considering semantically equivalent expressions. For example, the terms $(\lambda x, y.\ x - y)$, $(\lambda a, b.\ a - b)$, and $(\lambda y, x.\ y - x)$ are all capable of expressing exactly the same computations, so there should be a single canonical way of building this family of terms.

An important aspect of our canonicalization of semantically equivalent expressions is to enforce that *every term constructed during search has no free variables* (but terms may include variables referring to the inputs given in the I/O examples). Enforcing this property means that we can treat every term we build during search as an ordinary program, and run it on different inputs to probe its input-output behavior. However, the most straightforward way of building lambda terms bottom-up violates this important property. Consider the term $\mathtt{t}_5 = (\lambda v_1.\ \mathrm{map}(\lambda u.\ v_1 + u^2, \mathrm{sort}(x)))$ whose tree structure is illustrated in Figure 3(a), and where $x$ refers to an input variable. This has subterms such as $\lambda u.\ v_1 + u^2$, where the variable $v_1$ occurs free. As $v_1$ is free, it is unclear what the semantics of this expression should be. Constructing the expression with $v_1$ free would also introduce a spurious redundancy with $\lambda v_1, u.\ v_1 + u^2$. During search, we would like to keep only a canonical version of

those two terms to better prune the search space, which in this case would be $\mathtt{t}_3 = \lambda v_1, v_2. \, v_1 + v_2^2$, which as shown in Figure 3(b) can be used to construct the larger term $\mathtt{t}_5$.

To build terms bottom-up while maintaining desired constraints such as that every intermediate program has no free variables, we define an operator for algebraically combining smaller terms to build larger programs. This operator, which we call MERGE, takes as input a primitive DSL function $f$ and a list of terms $a_1 \ldots a_K$ constructed earlier during search, and it returns a new term that applies $f$ to $a_1 \ldots a_K$. MERGE does extra bookkeeping to ensure that every function (both $f$ and any $a_k$ that are lambda terms) is called with the correct number of arguments, and that the final output of MERGE has no free variables. The function $f$ can be a higher-order function, such as $\mathrm{map}$, or a normal first-order function, such as addition and subtraction.

Additional arguments to MERGE specify how to unify lambda arguments (if any) that appear in $a_1 \ldots a_K$. To evaluate MERGE on $f$ and $a_1 \ldots a_K$, we first apply each lambda argument $a_k$ to a tuple of variable names, denoted $i_k$. This gives a name to each argument of each $a_k$. By reusing the same name across different $a_k$'s, the same variable can be shared across arguments. For instance, if the function $f$ is multiplication ($\times$), and we are merging with the arguments $a_1 = (\lambda v. \, v + 1)$ and $a_2 = (\lambda v. \, v - 1)$, then we can share the variable by setting $i_1 = i_2 = [v_1]$, giving $(\lambda v. \, v + 1)(v_1) \times (\lambda v. \, v - 1)(v_1) = (v_1 + 1) \times (v_1 - 1)$. Alternatively, if we set $i_1 = [v_1]$ and $i_2 = [v_2]$, then merging gives a different program, $(v_1 + 1) \times (v_2 - 1)$. These tuples of variable names, $\{i_k\}_{k=1}^{K}$, are also input to MERGE, because they determine how variable names are unified across arguments. Finally MERGE runs $f$ on the arguments $\{a_k(i_k)\}_{k=1}^{K}$, pads the expression with lambdas at the beginning to bind any new free variables, and lastly canonicalizes variable naming and variable ordering.

Special care is needed for the arguments of higher-order functions like $\mathrm{map}$, which themselves need to be functions. So far, each argument $a_k(i_k)$ evaluates to a concrete value, not a function. To handle higher-order functions, MERGE automatically adds extra lambdas to each function-valued argument. These extra lambdas have variables denoted $u_1, u_2, \ldots$. All other variables used to unify variables across arguments are denoted $v_1, v_2, \ldots$. Although this may seem ad-hoc at first, this convention actually corresponds to a well-known canonicalization of typed lambda forms known as $\eta$-long normal form [28]. Putting everything together, MERGE is defined as

$$\mathrm{MERGE}\left(f, a_1, i_1, a_2, i_2, \ldots\right) = \lambda v_1 v_2 \ldots . \, f\left(\lambda u_1 u_2 \ldots u_{\ell_1}. \, a_1(i_1), \lambda u_1 u_2 \ldots u_{\ell_2}. \, a_2(i_2), \ldots\right)$$

$$\text{where } \{v_1, v_2, \ldots\} = \bigcup_k i_k - \{u_j\}_{j=1}^{\max\{\ell_1, \ell_2, \ldots\}} \tag{1}$$

where $\ell_k$ is the arity of the $k^{\text{th}}$ argument to $f$. For example, the higher-order function $\mathrm{map}$ first takes a function expecting one argument, so $\ell_1 = 1$, followed by a list (not a function) expecting no arguments, so $\ell_2 = 0$. We also enforce that $|i_k| = \mathrm{arity}(a_k)$. The MERGE operation is complete, in the sense that we can use it to generate any PBE solution within the DSL (see Appendix A).

Fundamentally, our neural model predicts the inputs to MERGE. It scans through its different operators (functions $f$), and for each operator, generates arguments $a_k$ by pointing into the set of explored values, and variables $i_k$ by emitting special tokens corresponding to $v_1, u_1, v_2, u_2$, etc. Figure 3(b) and (c) illustrate how a nontrivial lambda expression is built step by step, with the tokens emitted by the neural model highlighted in blue. In this figure, each call to MERGE in the third column returns the term that appears in the middle column, e.g., MERGE(square, $v_1$, [ ]) returns $\lambda v_1$. square($v_1$) and so on. Critically, MERGE makes sure that (1) every intermediate term that the model builds along the way can be evaluated so that the neural model can inspect its progress, and also (2) intermediate terms are generated with *every* function application, giving fine-grained feedback to the model.

We define the *weight* of an expression to be the number of nodes in its tree representation using the MERGE operator. More specifically, atomic terms like variable names and literal constants have weight 1, and a term constructed with MERGE has weight 1 (for the operation) plus the sum of the weights of all terms and variables in the MERGE call. For example, in Figure 3(b), terms $\mathtt{t}_1$ through $\mathtt{t}_5$ have weights 1, 2, 5, 2, and 10 respectively.

## 3.2 Learning over $\lambda$-Expressions

One core technical challenge is how to encode lambda expressions to allow neural models to reason about them. In LAMBDABEAM we solve this by constructing a new generalization of property

signatures which is designed to represent lambda expressions and non-lambda expressions using similar sets of properties.

Non-lambda expressions can be evaluated to produce a single result per I/O example. However, we cannot evaluate lambda expressions in the same way, because we do not know how the lambda expression will be used in the eventual solution, so we do not know what its arguments will be. For instance, if $x$ is an input list, the expressions $\text{zipwith}(\lambda u_1, u_2.\ u_1 + u_2, x, \text{sort}(x))$ and $\text{scanl1}(\lambda u_1, u_2.\ u_1 + u_2, x)$ use the same lambda expression $\lambda u_1, u_2.\ u_1 + u_2$ but for different sets of $(u_1, u_2)$ arguments. Thus, in order to describe the lambda expression's execution behavior, we run the lambda on a fixed set of *canonical argument tuples* based on the number of arguments and their types. These argument tuples are held constant across training and evaluation so that the model can learn from consistent signals.

In our experiments, we hardcoded the canonical argument tuples without changing them afterward, trying to cover common values and a variety of scenarios. For instance, our integer arguments include those between $-3$ and $5$ inclusive, and other integers with varying combinations of magnitude, sign, even/odd parity, primality, and so on. There is also a tradeoff in the number of canonical argument tuples, where having more leads to finer-grained execution information but more time spent running lambdas during search. In our experiments, we use 16 canonical argument tuples for each combination of tuple length and argument types in our DSL. Note that the lambda expression can refer to inputs from the I/O examples. Instead of running the lambda on each argument tuple for each example, for efficiency, we run on each argument tuple once, under the context of one I/O example which is changed per argument tuple in a round-robin fashion.

To represent a lambda $f$, we evaluate it on each canonical argument tuple $t_i$ with I/O example $(I, O)$. First we evaluate $f$ on $t_i$, yielding a result $r_i = f(t_i, I)$. Then we use a signature of unary properties to describe $r_i$. Second, we use a signature of binary properties to compare $r_i$ and $O$; intuitively, this helps the model learn about what work remains. Similarly, we use the binary properties to compare $r_i$ and $t_{ij}$, for each argument $t_{ij} \in t_i$. By concatenating these, we obtain a single property vector for each tuple $t_i$. Finally, we then reduce the property vectors across the runs of the lambda, i.e., computing for each property the fraction of runs where it is applicable and the fraction of applicable runs where it is True. Encoding non-lambda expressions is similar, except that we use I/O examples $(I_i, O_i)$ in place of the canonical tuples. Note that the reduced property signatures for lambda and non-lambda expressions have different formats and different lengths, and hence they are embedded by separate parts of the neural model (Section 3.3).

The properties used in our property signatures come from combinatorially combining hand-designed features as "building blocks" to create a rich space of properties that describe individual objects as well as comparisons between two objects. Appendix B contains more details.

### 3.3 LAMBDABEAM Model and Search

To guide the bottom-up search over lambda and non-lambda expressions, we generally follow the design of the neural policy network in CROSSBEAM [33], with the following major changes:

**Value Module**  We maintain a set of explored values $\mathcal{S}$ which contains variable tokens for constructing lambdas, lambda expressions, and non-lambda expressions including constants and input variables. The Value Module embeds each element of $\mathcal{S}$ forming a matrix of embeddings $E_{\mathcal{S}} \in \mathbb{R}^{|\mathcal{S}| \times d}$. Elements of $\mathcal{S}$ are embedded as follows. A variable token is embedded as a vector of trainable parameters. Note that the set of such variable tokens is fixed and determined by the DSL.[3] A lambda expression is embedded by $\mathbf{s} + \mathbf{z}$, where $\mathbf{s}$ is the property signature of this lambda function encoded by an MLP, and $\mathbf{z}$ is an embedding of the weight of this value. Non-lambda expressions are embedded like lambda expressions except using a different MLP to encode their property signatures.

**Argument Selector Module**  Given an operator $op$, we use an operator-specific $\text{LSTM}_{op}$ to select the arguments using a pointer mechanism [35] from the encoded value matrix $E_{\mathcal{S}}$, in an autoregressive way. In addition to selecting $\text{arity}(op)$ arguments, for an argument $a_k$ that is a lambda expression, we also need to predict the variables $i_k$ as required for MERGE, where $i_k$ is a tuple of $\text{arity}(a_k)$ variable tokens. All of the $a_k$ and $i_k$ predictions are done as a single autoregressive sequence. Furthermore,

---

[3]The higher-order functions in our DSL expect lambdas with at most 2 variables. Thus, it is unnecessary to create lambdas with 3+ variables, so the only variables needed for MERGE are $v_1$, $v_2$, $u_1$, and $u_2$.

for convenience we predict all of the $a_k$ arguments first, followed by the $i_k$ variable tuples which are constrained (via masking and padding) to include exactly $\text{arity}(a_k)$ variable tokens.

**Search with Restarts** We also change the inference time search procedure. Recall that CROSS-BEAM uses random sampling during evaluation, making the search nondeterministic. In LAMBDA-BEAM, instead of performing one synthesis search until timeout, we restart the search from scratch whenever the search has run for a certain amount of time without finding a solution. Even though this discards work done in previous searches, in practice this helps LAMBDABEAM solve more tasks because it may be otherwise difficult to recover from exploring the wrong search direction.

# 4 Experiments

In this section, we experimentally evaluate the effectiveness of LAMBDABEAM, comparing to prior neural and symbolic approaches in an integer list manipulation domain.

## 4.1 Integer List Manipulation DSL

To measure a synthesizer's ability to create and use lambda expressions, we create a domain-specific language (DSL) that emphasizes lambda expressions and higher-order functions. Specifically, the DSL from DeepCoder [3] includes higher-order functions and has been used in subsequent work [37, 34]. However, DeepCoder's DSL only contains a hardcoded set of lambda functions and is not expressive enough to fully exercise LAMBDABEAM's ability to create arbitrary lambda expressions. Therefore, we extend DeepCoder's DSL by allowing lambda expressions to include arbitrary compositions of DSL operations, and replacing the hardcoded lambda functions with DSL operations and literal constants that enable a superset of the original functionality. For example, DeepCoder's original DSL includes hardcoded lambdas such as $(\lambda x.\ x - 1)$, $(\lambda x, y.\ x - y)$, and $(\lambda x, y.\ \max\{x, y\})$. By introducing first-order functions including subtract and max, and constant literals including 0 and 1, we can create the hardcoded lambdas as well as lambdas like $(\lambda x. \max\{x, 0 - x\})$ that were not possible in the original DeepCoder DSL. Additionally, we add a new if-then-else operation, which further enriches our space of possible programs. The full DSL contains 23 first-order operations, 5 higher-order operations, and 6 integer literals, described fully in Appendix C. In our DSL, all integers are in the range $[-256, 255]$ as in DeepCoder, and lists have lengths in the range $[0, 10]$.

## 4.2 Experimental Setup

**Training Data** Similar to previous works including CROSSBEAM, we create synthetic training data by generating random tasks within our DSL. This is done by performing exhaustive bottom-up searches starting from random inputs and enumerating programs in order of increasing weight, and then sampling a subset of the resulting programs to serve as training tasks. Each task has between 2 and 5 I/O examples and between 1 and 3 input variables, and we sample tasks such that approximately 80% of them use lambdas in the ground-truth program. We used a time limit of 1 hour per data generation search (reaching programs of weight at most 12), sampling up to 1600 tasks per search, and performing enough searches parallelized across cloud CPU workers such that training uses less than 1 epoch over the dataset. We furthermore removed from the training dataset all programs that would solve any of our 200 evaluation tasks, described below.

**Evaluation Tasks** For evaluation, we use 100 handwritten evaluation tasks plus 100 synthetically generated tasks, with a time limit of 10 minutes per task. The handwritten tasks include all 9 "example programs" from Appendix A of the DeepCoder paper [3], plus other tasks that we created from scratch by brainstorming many natural but varied tasks that we could solve using our DSL (near the end of this process, it became quite difficult to come up with new tasks that were not merely slight variations of existing ones). Handwritten tasks include between 1 and 3 input variables, 3 I/O examples if the output is a list or 5 I/O examples if the output is an integer, and a handwritten ground-truth solution that has minimal weight to our knowledge. When creating I/O examples, we aimed to make the examples informative but reasonably succinct with lists of length at most 10. Every DSL operation is used in the solution for at least 4 handwritten tasks, and every higher-order operation is used in at least 10. For the 100 synthetic tasks, we sampled distinct random programs using the same procedure as for generating training data, except also enforcing that we have exactly

10 tasks of each weight between 3 and 12 inclusive. Appendix D contains example handwritten and synthetic tasks, along with LAMBDABEAM's solutions for them.

**Approaches** Our experiments compare several approaches:

1. LAMBDABEAM with random restarts: We trained the LAMBDABEAM model using on-policy training as in CROSSBEAM. The model has about 13 million trainable parameters and was trained on about 6.5 million tasks, which took about a week of training using 8 V100 GPUs. See Appendix E for more details on the model architecture and training. During evaluation, we use only 1 V100 GPU and perform random restarts every 6 seconds on the handwritten tasks, or every 30 seconds on the synthetic tasks, both chosen from a coarse search over restart frequencies. We run this approach for 5 trials using different randomness for the UniqueRandomizer sampling method carried over from CROSSBEAM.

2. LAMBDABEAM without random restarts: We use the LAMBDABEAM approach without random restarts as an ablation, also for 5 trials with different randomness for UniqueRandomizer sampling.

3. Enumeration: We run the same exhaustive bottom-up enumerative synthesis algorithm that was used to create the training data. We use 5 trials with different random orderings of DSL operations, which changes the enumeration order of programs with the same weight.

4. RobustFill [12]: This approach treats the synthesis problem as a sequence to sequence prediction task from the I/O examples to the program tokens. Specifically, we train a plain 3-layer LSTM-based encoder-decoder model on our synthetic training dataset, using approximately the same number of trainable parameters and training tasks as for the LAMBDABEAM model. We get model predictions via a single beam search of size 65536 which nearly exhausts the GPU memory and evaluate all resulting programs on the I/O examples. Since the beam search is deterministic, we perform 5 trials by re-training the model with different initializations.

5. $\lambda^2$ [15]: This is a state-of-the-art symbolic program synthesizer that handles lambda functions and higher-order functions. We implemented our DSL within the $\lambda^2$ framework (using the more extensible version provided by the $\lambda^2$ authors). $\lambda^2$ is deterministic so we only use 1 trial.

6. Python-Finetuned LLM: We try asking a pretrained large language model (LLM) to solve our evaluation tasks using Python code. Specifically, we use PaLM 62B that was trained for longer as described in Appendix F of Chowdhery et al. [9], with further fine-tuning on general Python code. The prompt contains natural language instructions and 2 examples of an I/O specification followed by Python code that solves the task (for 2 new tasks), and then the I/O specification of the evaluation task we wish to solve.[4] We repeatedly draw batches of 16 independent samples with temperature sampling and run those programs on the I/O examples, until a solution is found or timeout is reached. We ran the LLM using 16 accelerators so this approach uses significantly more compute than the others over the same time limit. We repeat for 3 trials with different randomness for temperature sampling.

## 4.3 Results

Figure 4 plots the synthesis performance of the various methods over time. Notably, LAMBDABEAM with restarts is the best approach for both handwritten and synthetic tasks. The gap is wider on the handwritten tasks where LAMBDABEAM with restarts solves 67.2 out of 100 tasks on average, which is 24% more tasks than the next best method $\lambda^2$. Figure 5 plots the various approaches' success rates for different task weights, which can be used as a measure of task difficulty. As expected, we observe that all methods perform worse on harder tasks with larger weight, but that LAMBDABEAM with restarts generally achieves higher success rates on the difficult tasks compared to other methods. This means that our approach scales better to larger programs compared to the other methods, except the LLM which has a more constant but lower success rate overall.[5]

---

[4]In preliminary experiments, we found that providing 3 few-shot examples in the prompt led to slower sampling without much change in program quality. On the other hand, using only 1 example led to noticeably worse program quality. We also tried asking for programs within our DSL via few-shot examples, but this was not as successful because the LLM was not trained on our DSL.

[5]The LLM predicts Python code instead of using our DSL, so the weight according to our DSL is an inaccurate measure of the complexity of the corresponding Python code. Furthermore, difficulty for the LLM is more closely correlated with how "natural" the task is, i.e., its similarity to programs in the LLM's training data.

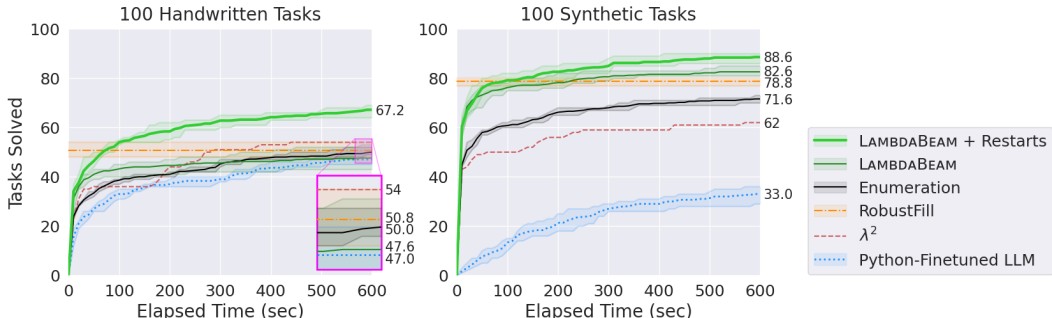

Figure 4: Synthesis results over time for various methods on the handwritten and synthetic evaluation tasks. Shaded areas represent the minimum and maximum across trials for that method.

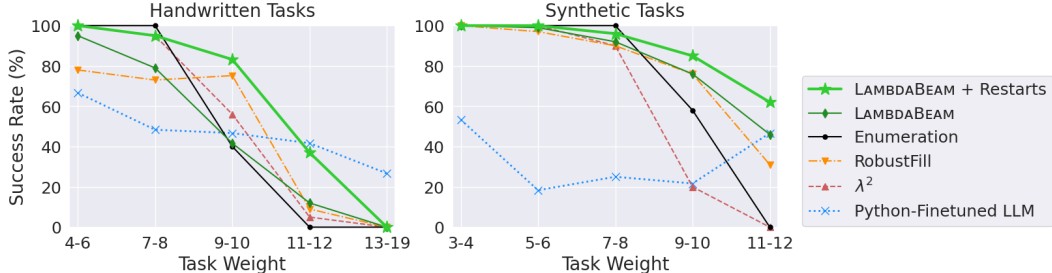

Figure 5: Success rates of various methods broken down by the task weight (i.e., the smallest known weight of a solution). Task weights are bucketed such that each group contains at least 15 tasks.

Running LAMBDABEAM with random restarts helps overall but more so for the handwritten tasks. We believe this is because the synthetic evaluation tasks have the same distribution as the training tasks while the handwritten tasks are different. So, for the handwritten tasks, exploring the wrong part of the search space early on might cause further mistakes that lead the search astray, while the model may be better trained to stay on track for the synthetic tasks. This would also explain why more frequent restarts are effective for the handwritten tasks. We note that random restarts would not be possible for $\lambda^2$, enumeration, or RobustFill's beam search, and would not help the LLM where each sample is already independent.

We also identify *false positives* by running solutions on 2 held-out test cases per task, generated mostly synthetically with some manual editing. The results are in Figure 6, showing that LAMBDABEAM with restarts has the highest number of true positive solutions on handwritten tasks by a margin of nearly 8 tasks, while barely losing to Enumeration on synthetic tasks.[6] While symbolic approaches (Enumeration and $\lambda^2$) have fewer false positives due to focusing on small solutions, we observe that LAMBDABEAM has the fewest false positives among the neural approaches. The LLM produces many false positives on the synthetic tasks where the ground-truth solutions are less similar to programs seen during its training, and in fact many of its false positive solutions are if-elif-else chains that hardcode the examples in some way (which is feasible to implement in Python but not in our DSL). Finally, we note that some false positive solutions could be transformed into true positives with a postprocessing step, e.g., one that attempts to simplify or minimize subtrees of the solution. In this sense, false positive solutions may still be useful for synthesis, and LAMBDABEAM with restarts achieves the highest number of total positive solutions by a wide margin.

Appendix F contains analysis showing some of the differences in distributions between the handwritten and synthetic evaluation tasks, which helps to contextualize the experimental results. For example, lambda expressions are used in 85% of the handwritten tasks but only 53% of the synthetic tasks. The median task weight is 9 for handwritten tasks and only 7.5 for synthetic tasks. These comparisons suggest that the handwritten tasks are harder than the synthetic tasks on average, which

---

[6]The synthetic tasks have randomly-generated I/O examples that are overall less informative than those in the handwritten tasks, and the "correct" solution is not chosen to be *natural* but rather is one with minimal weight by construction. Enumeration has an unfair advantage here, being the only method in our comparison that is *guaranteed* to find a minimal weight solution.

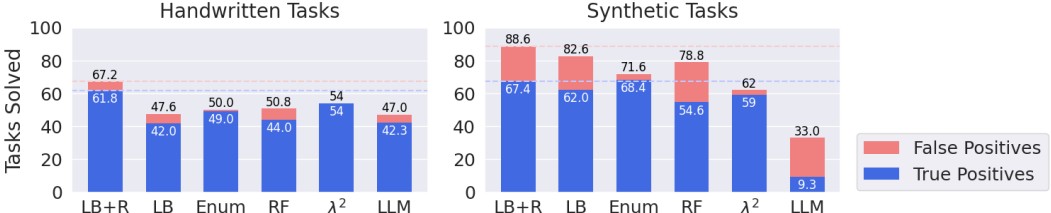

Figure 6: True positive versus false positive solutions measured using held-out test cases.

is also reflected in the overall performance in Figure 4. We observe that LAMBDABEAM achieves a greater performance gap over the other approaches on the handwritten tasks versus on the synthetic tasks, which is a promising trend because the handwritten tasks are both harder and more natural.

Although LAMBDABEAM resolves CROSSBEAM's limitations of not handling lambdas or looping computations, some other limitations are carried over. On-policy training is slow due to performing search during training, but this could be addressed with an initial training phase of off-policy teacher forcing. At evaluation time, even with UniqueRandomizer sampling to avoid duplicate argument lists within one sampling phase, our approach still encounters many duplicate values across sampling phases and across restarts. Finally, our DSL is small compared to general programming languages.

## 5 Related Work

Machine learning for program synthesis has been an active area [17, 18, 1]. Within programming by example, deep learning architectures for sequences, like LSTMs and Transformers, have been particularly effective [12]. Our work builds upon CROSSBEAM [33], which itself combines three lines of research in program synthesis. The first are learned search strategies for program synthesis, that is, using a learned policy or value function to guide search [36, 20, 13], or multi-level strategies that combine the results of search over different spaces [25, 22, 19, 34]. The second are *execution-guided neural synthesis* methods, which guide the search over partial programs by evaluating them [37, 13, 7, 27, 8]. Finally, CROSSBEAM's use of imitation learning to train the policy is inspired by work in learning to search [11, 29, 5] and beam-aware training [23, 24].

In contrast, we are unaware of previous work that synthesizes helper functions, such as lambda functions, during neural program synthesis. The original DeepCoder DSL contains only a small set of predefined lambda functions. Even within symbolic program synthesis, $\lambda^2$ is one of the few examples of work that synthesizes lambda expressions [15]. To control the size of the search space, $\lambda^2$ employs type-directed synthesis, but we handle more general domains where the type system is not informative enough to reduce the search space sufficiently. DreamCoder [14] can also infer ad-hoc helper functions like $\lambda^2$, but its neural network provides no fine-grained guidance on how to compose those lambdas. Because DreamCoder is an algorithm for enriching an impoverished DSL to improve a neurally-guided program search, one could combine DreamCoder's DSL enrichment process with LAMBDABEAM's search strategy. Other work reuses fragments of code from partially-correct solutions [30, 34], but these are executable portions of straightline code, not lambda functions.

Our integer manipulation domain is inspired by DeepCoder [3] and subsequent work [37, 34].

## 6 Conclusion

We introduced the first neural search method for programming by example that is able to synthesize intermediate helper functions (lambdas) by resolving two key difficulties. First, we algebraically represent lambda expressions in a canonical way and construct new lambdas with the MERGE operator that enforces desirable representational constraints. Second, we encode arbitrary lambda functions as inputs to a neural network by using property signatures to analyze the lambda's execution semantics. With these innovations, LAMBDABEAM learns a neural policy to drive a bottom-up search over programs. We experimentally show that LAMBDABEAM outperforms symbolic search, a sequence model, and a pretrained code LLM with 62 billion parameters.

**Acknowledgments**

The authors would like to thank Henryk Michalewski for his thoughtful ideas, and Christian Walder, Rif Saurous, and the anonymous reviewers for their helpful comments.

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

# Appendices

## A Completeness of MERGE

The MERGE operation is complete in the sense that it can generate all possible solutions in the domain-specific language (DSL) for a programming-by-example (PBE) problem.

We formalize our DSL in a subset of the lambda calculus. Let $\mathcal{X} = \{x_1, \ldots, x_m\}$ be the set of input variables for the PBE task, $\mathcal{V}$ be a countable set of variables that is disjoint from $\mathcal{X}$, $\mathcal{F}$ be the set of primitive functions in the DSL, and $\mathcal{C}$ be a set of constants in the DSL. Then, our lambda calculus is:

$$T ::= x \mid v \mid c \mid f(\mathbf{t}_1, \ldots, \mathbf{t}_k) \mid \lambda v_1...v_n.\, \mathbf{t}$$
$$\text{for } x \in \mathcal{X}, c \in \mathcal{C}, f \in \mathcal{F},$$
$$v, v_1, \ldots, v_n \in \mathcal{V},$$
$$\mathbf{t}, \mathbf{t}_1, \ldots, \mathbf{t}_k \in T.$$

Let $M$ be the set of terms obtainable by repeatedly applying MERGE (including the initial terms usable by MERGE):

$$M ::= x \mid \lambda v.\, v \mid c \mid \text{MERGE}(f, \mathbf{a}_1, i_1, \ldots, \mathbf{a}_k, i_k)$$
$$\text{for } x \in \mathcal{X}, v \in \mathcal{V}, c \in \mathcal{C}, f \in \mathcal{F},$$
$$\mathbf{a}_1, \ldots, \mathbf{a}_k \in M,$$
$$i_1, \ldots, i_k \in \mathcal{V}^*.$$

We restate the definition

$$\text{MERGE}\left(f, \mathbf{a}_1, i_1, \mathbf{a}_2, i_2, \ldots\right) = \lambda v_1 v_2.... \, f\left(\lambda u_1 u_2...u_{\ell_1}.\, \mathbf{a}_1(i_1), \lambda u_1 u_2...u_{\ell_2}.\, \mathbf{a}_2(i_2), \ldots\right)$$
$$\text{where } \{v_1, v_2, \ldots\} = \bigcup_k i_k - \{u_j\}_{j=1}^{\max\{\ell_1, \ell_2, \ldots\}}$$

From this definition, it is clear that $M \subseteq T$, i.e., MERGE is closed within the lambda calculus. However, $M \neq T$ because MERGE imposes certain constraints, e.g., $\lambda v.\, x$ is in $T$ but cannot be constructed by MERGE. To precisely describe the constraints resulting from MERGE, we introduce the following definitions:

- Terms $x \in \mathcal{X}$, $v \in \mathcal{V}$, and $c \in \mathcal{C}$ are *atomic*.
- A term $\mathbf{s} = \lambda v_1...v_n.\, \mathbf{t}$ (possibly with $n = 0$ such that $\mathbf{s}$ is not a lambda expression) *has exact lambda variables* if $\text{FreeVars}(\mathbf{t}) - \mathcal{X} = \{v_1, \ldots, v_n\}$. Note that $\mathbf{s}$ having exact lambda variables implies that $\text{FreeVars}(\mathbf{s}) \subseteq \mathcal{X}$.
- A term *typechecks* if every function application has the correct arity for every argument, e.g., $\text{Map}(\mathbf{t}_1, \mathbf{t}_2)$ expects $\mathbf{t}_1$ to have arity 1, while $\mathbf{t}_2$ should have arity 0.

We now let $S = \{\mathbf{s} \in T \mid \mathbf{s} \text{ has exact lambda variables and typechecks}\}$.

The completeness of MERGE, in the sense that it can generate all solutions $\mathbf{s}$ to PBE problems (once the input variables $x_1, \ldots, x_m$ are bound), follows from the two claims below.

**Claim 1.** If $\mathbf{p} = \lambda x_1...x_m.\, \mathbf{s}$ is a solution to a PBE problem, such that $\mathbf{p}(x_1, \ldots, x_m) = y$ for all I/O examples $(x_1, \ldots, x_m) \to y$ in the PBE specification, then $\mathbf{s} \in S$. That is, $S$ is broad enough to cover all solutions to PBE problems.

*Proof.* To ensure that $\mathbf{p} = \lambda x_1...x_m.\, \mathbf{s}$ is a valid solution program, we must have $\text{FreeVars}(\mathbf{s}) - \mathcal{X} = \emptyset$ (so there are no unbound variables), $\mathbf{s}$ must have arity 0 (since all inputs $x_1, \ldots, x_m$ are already bound), and $\mathbf{s}$ must typecheck to avoid runtime errors. Together, these imply that $\mathbf{s} \in S$. $\qquad\square$

**Claim 2.** $S \subseteq M$. That is, MERGE can create any term in $S$, including all solutions to PBE problems.

*Proof.* Let $\mathbf{s}$ be any term in $S$. We will proceed by induction on the depth of $\mathbf{s}$.

As the base case, if $\mathbf{s}$ is atomic, then $\mathbf{s} = x$ or $\mathbf{s} = c$, so $\mathbf{s}$ is immediately in $M$. Note that $\mathbf{s}$ cannot be $v$ because $v$ does not have exact lambda variables.

Then, we assume the inductive hypothesis that any term in $S$, with depth less than that of $\mathbf{s}$, is in $M$. There are two inductive cases to consider: either $\mathbf{s} = \lambda v_1...v_n.\ f(\mathbf{t}_1, \ldots, \mathbf{t}_k)$ where $n$ might be 0, or $\mathbf{s} = \lambda v_1...v_n.\ \mathbf{t}$ where $n > 0$ and $\mathbf{t}$ is atomic. Because $\mathbf{s}$ has exact lambda variables, the latter scenario is only possible if $\mathbf{s} = \lambda v.\ v$, which is immediately in $M$.

The remaining case is $\mathbf{s} = \lambda v_1...v_n.\ f(\mathbf{t}_1, \ldots, \mathbf{t}_k)$. Consider any $\mathbf{t}_j$ for $1 \leq j \leq k$. We will construct $\mathbf{a}_j$ and $i_j$ such that $\mathrm{MERGE}(f, \mathbf{a}_1, i_1, \ldots, \mathbf{a}_k, i_k) = \mathbf{s}$.

- If $\mathbf{t}_j$ is atomic, then either $\mathbf{t}_j = x$, $\mathbf{t}_j = v$, or $\mathbf{t}_j = c$. If $\mathbf{t}_j = v$, then set $\mathbf{a}_j = \lambda v.\ v$ and $i_j = [v]$; otherwise, set $\mathbf{a}_j = \mathbf{t}_j$ and $i_j = [\,]$, the empty tuple. In each case, $\mathbf{a}_j \in M$. Because $\mathbf{s}$ typechecks, we know that the $j$-th argument to $f$ expects arity 0, so in the $\mathrm{MERGE}$ definition, $\ell_j = 0$ and thus the $j$-th argument to $f$ expands to $\mathbf{a}_j(i_j) = \mathbf{t}_j$ in each case.[7]

- If $\mathbf{t}_j$ is not atomic, then let $\mathbf{t}_j = \lambda u_1...u_{\ell_j}.\ \mathbf{r}$, where $\ell_j$ is the expected arity of the $j$-th argument to $f$ (because $\mathbf{s}$ typechecks). Let $\{v_1', \ldots, v_d'\} = \mathrm{FreeVars}(\mathbf{r}) - \mathcal{X}$. Set $\mathbf{a}_j = \lambda v_1'...v_d'.\ \mathbf{r}$ and $i_j = [v_1', \ldots, v_d']$, so that when applying $\mathrm{MERGE}$, the $j$-th argument to $f$ expands to $\lambda u_1...u_{\ell_j}.\ \mathbf{a}_j(i_j) = \lambda u_1...u_{\ell_j}.\ \mathbf{r} = \mathbf{t}_j$. Furthermore, $\mathbf{a}_j$ has exact lambda variables by construction, and it typechecks because $\mathbf{r}$ typechecks, so $\mathbf{a}_j \in S$ and $\mathbf{a}_j \in M$ by the inductive hypothesis.

With these choices of $\mathbf{a}_j$ and $i_j$, when expanding $\mathrm{MERGE}(f, \mathbf{a}_1, i_1, \ldots, \mathbf{a}_k, i_k)$ according to the definition, the $j$-th argument to $f$ becomes $\mathbf{t}_j$, and the lambda variables $\bigcup_k i_k - \{u_j\}_{j=1}^{\max\{\ell_1, \ell_2, \ldots\}}$ are exactly $\mathrm{FreeVars}(f(\mathbf{t}_1, \ldots, \mathbf{t}_k)) = \{v_1, \ldots, v_n\}$ since $\mathbf{s}$ has exact lambda variables. Therefore, $\mathbf{s} = \lambda v_1...v_n.\ f(\mathbf{t}_1, \ldots, \mathbf{t}_k) = \mathrm{MERGE}(f, \mathbf{a}_1, i_1, \ldots, \mathbf{a}_k, i_k)$, so $\mathbf{s} \in M$. $\square$

# B  More Details on Property Signatures

Here we describe in more detail the properties we use to encode lambda and non-lambda values. We define the following helper functions to organize the properties.

First, $\mathrm{TypeProperties}(x)$ represents the type of $x$ as a boolean one-hot list. In our DSL, this returns a tuple of 5 booleans, representing whether $x$ is a lambda, boolean, int, list, or None (which is used to indicate an error, e.g., signaling that an index is out of bounds).

Next, we define $\mathrm{BasicProperties}(x)$ to evaluate hand-designed "basic" properties that describe objects of each different type in the DSL. This returns a fixed-length vector of property results (each being True, False, or N/A). Note that, if $x$ has type $\tau$, then all properties for type $\tau$ evaluate to True or False, while all properties for all other types $\tau' \neq \tau$ evaluate to N/A. For our DSL, we use the following basic properties:

- For boolean $x$: $x$ itself.
- For integer $x$: whether $x$ equals $-1$, 0, 1, and 2; whether $x$ is positive and negative; whether $x$ is even; whether $x$ is 0 and 1 modulo 3; and whether $|x|$ is less than 5, 10, 20, 35, 50, 75, and 100.
- For list $x$: whether $x$ is sorted, whether $x$ is sorted in reverse, and whether $x$ contains all unique elements.

Then, $\mathrm{Relevant}(x)$ returns related objects that are relevant to understanding $x$. For our DSL, this is only $x$ itself for integer and boolean $x$, but for list $x$, the "relevant" objects are: $x$ itself; the length of $x$; the number of distinct elements in $x$; the max, min, range, and sum of $x$; and the first and last elements of $x$ (defaulting to 0 if $x$ is empty).

This culminates in $\mathrm{ObjectSignature}(x)$ which takes a single DSL object $x$ and returns a fixed-length vector of property results, containing $\mathrm{TypeProperties}(x)$ followed by $\mathrm{BasicProperties}(r)$ for each $r \in \mathrm{Relevant}(x)$. For example, these properties include "assuming $x$ is an int, is $x$ even?" (a basic

---

[7] In practice for simplicity, when $\mathbf{t}_j = v$, we simply set $\mathbf{a}_j = \mathbf{t}_j$ and $i_j = [\,]$ as in the other cases, even though $\mathbf{a}_j = v$ does not have exact lambda variables.

property applied to $x$) as well as "assuming $x$ is a list, are there an even number of elements in $x$?" (a basic property applied to an object relevant to $x$). By applying basic properties to relevant objects in this compositional way, we reduce the effort needed to specify a large number of properties.

We furthermore encode comparisons between two objects. We define $\mathrm{ComparisonProperties}(x, y)$ which evaluates hand-designed properties for comparing two objects $x$ and $y$ of the same type, for each different type in the DSL. This returns a fixed-length vector of property results, where a property for comparing type $\tau$ evaluates to N/A if $x$ and $y$ are not of type $\tau$. For our DSL, we use the following comparison properties:

- For boolean $x$ and $y$: whether $x = y$.
- For integer $x$ and $y$: whether $x = y$, $x < y$, and $x > y$; whether $x$ is a factor of $y$ and vice versa; and whether $|x - y|$ is less than 2, 5, 10, and 20.
- For list $x$ and $y$: whether $x = y$; whether $x$ is longer, shorter, or equal length compared to $y$; whether the lengths differ by at most 1; whether all $x_i < y_i$ for $x_i, y_i \in \mathrm{zip}(x, y)$ and similarly for other comparisons $\leq, >, \geq, =$, and $\neq$; whether $x$ and $y$ contain the same set of elements; and whether $x$ contains a subset of elements compared to $y$ and vice versa.

These properties are used in $\mathrm{ComparisonSignature}(x, y)$ which computes a fixed-length list of property results for any two DSL objects $x$ and $y$ of any type, containing $\mathrm{ComparisonProperties}(r_x, y)$ for all $r_x \in \mathrm{Relevant}(x)$ where $\mathrm{type}(r_x) = \mathrm{type}(y)$, and $\mathrm{ComparisonProperties}(x, r_y)$ for all $r_y \in \mathrm{Relevant}(y)$ where $\mathrm{type}(r_y) = \mathrm{type}(x)$. Thus, "assuming $x$ is an int and $y$ is a list, is $x$ a factor of the length of $y$?" is one resulting property. As usual, if $x$ and $y$ do not match the types assumed by the property, then the property evaluates to N/A.

In the I/O Module of the neural policy (as in CROSSBEAM [33]), we use property signatures to encode a set of I/O examples. For each example $(\{I_1, \ldots, I_n\}, O)$ we concatenate $\mathrm{ObjectSignature}(O)$ with $\mathrm{ObjectSignature}(I_i)$ and $\mathrm{ComparisonSignature}(I_i, O)$ for all $1 \leq i \leq n$. Then, we reduce these across I/O examples, computing for each property the fraction of examples where it is applicable (not N/A), and the fraction of examples where it is True among those where it is applicable (defaulting to 0.5 if it is N/A for all examples).

In the Value Module of the neural policy, we use property signatures to encode a value (lambda or non-lambda expression) that was found during search. To encode a lambda expression, we run it on canonical input tuples as described in Section 3.2. For each run of the lambda on canonical input tuple $t_i = (t_{i,1}, \ldots, t_{i,m})$ using an I/O example $(I, O)$ where the lambda evaluates to a result $r_i$, we concatenate $\mathrm{ObjectSignature}(r_i)$, $\mathrm{ComparisonSignature}(r_i, O)$, and $\mathrm{ComparisonSignature}(t_{i,j}, r_i)$ for all $1 \leq j \leq m$, and then reduce these across the runs of the lambda. To encode a non-lambda expression during search, for each I/O example $(I, O)$ where the expression evaluates to a result $r$, we concatenate $\mathrm{ObjectSignature}(r)$ with $\mathrm{ComparisonSignature}(r, O)$, and then reduce these across I/O examples. Note that the signatures for values found during search do not contain comparisons to the I/O example inputs, because what ultimately matters is whether the value is useful for creating the *output* later, not how the value was created from the inputs.

In our implementation, encoding the set of I/O examples results in a property signature of length 1230, encoding a lambda expression results in a property signature of length 558, and encoding a non-lambda expression results in a property signature of length 359.

## C  Extension of the DeepCoder DSL

As mentioned in Section 4.1, we extended the DSL from DeepCoder [3]. Atomic terms in the DSL include variable names and the constant literals $-1$, $0$, $1$, $2$, $3$, and $4$. The DSL contains 23 first-order and 5 higher-order operations, listed below with type annotations and Python implementations:

```python
# 23 first-order operations

def Add(x: int, y: int) -> int:
  return x + y

def Subtract(x: int, y: int) -> int:
  return x - y

def Multiply(x: int, y: int) -> int:
  return x * y
```

```python
def IntDivide(x: int, y: int) -> int:
  return x // y

def Square(x: int) -> int:
  return x ** 2

def Min(x: int, y: int) -> int:
  return min(x, y)

def Max(x: int, y: int) -> int:
  return max(x, y)

def Greater(x: int, y: int) -> bool:
  return x > y

def Less(x: int, y: int) -> bool:
  return x < y

def Equal(x: int, y: int) -> bool:
  return x == y

def IsEven(x: int) -> bool:
  return x % 2 == 0

def IsOdd(x: int) -> bool:
  return x % 2 != 0

def If(c: bool, x: int, y: int) -> int:
  return x if c else y

def Head(xs: list) -> int:
  return xs[0]

def Last(xs: list) -> int:
  return xs[-1]

def Take(n: int, xs: list) -> list:
  return xs[:n]

def Drop(n: int, xs: list) -> list:
  return xs[n:]

def Access(n: int, xs: list) -> int:
  return xs[n]

def Minimum(xs: list) -> int:
  return min(xs)

def Maximum(xs: list) -> int:
  return max(xs)

def Reverse(xs: list) -> list:
  return list(reversed(xs))

def Sort(xs: list) -> list:
  return sorted(xs)

def Sum(xs: list) -> int:
  return sum(xs)

# 5 higher-order operations

def Map(f: Callable[[int], int], xs: list) -> list:
  return [f(x) for x in xs]

def Filter(f: Callable[[int], bool], xs: list) -> list:
  return [x for x in xs if f(x)]

def Count(f: Callable[[int], bool], xs: list) -> int:
  return len([x for x in xs if f(x)])

def ZipWith(f: Callable[[int, int], int], xs: list, ys: list) -> list:
  return [f(x, y) for x, y in zip(xs, ys)]

def Scan1(f: Callable[[int, int], int], xs: list) -> list:
  ys = [xs[0]]
  for n in range(1, len(xs)):
    ys.append(f(ys[n-1], xs[n]))
  return ys
```

# D  Example Tasks

This section contains selected example problems from our 100 handwritten and 100 synthetic evaluation tasks. Each task is given a name for convenience purposes only, which is not used by any method in our experiments.

## D.1  Handwritten Task "`map:replace`"

This task has 3 inputs (`x`, `f`, and `r`), 3 examples demonstrating the task ("in `x`, find instances of `f` and replace them with `r`"), and a handwritten ground-truth solution using a relatively complicated lambda function:

```
inputs_dict = {
    'x': [[7, 2, 4, 6, 4, 2, 5],
          [-6, -3, 4, 3, -5, -3, 2, 1, 5],
          [18, 48, 27, 26, 27, 27, 28, 17, 27, 33]],
    'f': [4, -3, 27],
    'r': [-1, 7, 99],
}
outputs = [[7, 2, -1, 6, -1, 2, 5],
           [-6, 7, 4, 3, -5, 7, 2, 1, 5],
           [18, 48, 99, 26, 99, 99, 28, 17, 99, 33]]
solution = 'Map(lambda u1: If(Equal(u1, f), r, u1), x)'
```

In `inputs_dict`, each of the entries for `x`, `f`, and `r` is a list of length 3, which contains the input for each of the 3 examples. Similarly, `outputs` is a list containing the output for each example. `solution` is our handwritten solution.

LAMBDABEAM + Restarts finds the same solution of weight 10 in each of the 5 trials, taking a median time of 202 seconds:

```
Map(lambda u1: (lambda v1: If((lambda v1: Equal(f, v1))(v1), r, v1))(u1), x)
```

The solution looks complicated due to the MERGE operation causing lots of variable renames (i.e., $a_k(i_k)$ in the MERGE definition). We have implemented an algorithm to simplify the solution by statically resolving these renames. In this case, the solution simplifies to

```
Map(lambda u1: If(Equal(f, u1), r, u1), x)
```

which is essentially identical to the ground-truth solution.

## D.2  Handwritten Task "`multi:multiply_odds`"

This task has 1 input and uses multiple higher-order functions to compute a running product of only the odd elements:

```
inputs_dict = {
    'x': [[3, 5, 8, 2, 1],
          [5, 2, 1, 3, 3, 1, 4],
          [3, -4, -1, 8, 2, 0, -3, 0, 9, -1]],
}
outputs = [[3, 15, 15],
           [5, 5, 15, 45, 45],
           [3, -3, 9, 81, -81]]
solution = 'Scanl1(lambda u1, u2: Multiply(u1, u2), Filter(lambda u1: IsOdd(u1), x))'
```

In each of the 5 trials, LAMBDABEAM + Restarts finds the same solution of weight 11 that simplifies to the ground-truth solution, taking a median time of 75 seconds.

## D.3  Synthetic Task "`synthetic:weight_9_function_7`"

This task clips every element to the range $[0, 4]$:

```
inputs_dict = {
    'x1': [[-9, -2, -10, -6, 0, -10, -6, 3, 1],
           [-1, -5, 8, 5]]
}
outputs= [[0, 0, 0, 0, 0, 0, 0, 3, 1],
          [0, 0, 4, 4]]
solution = 'Map(lambda u1: Min(4, Max(0, u1)), x1)'
```

LAMBDABEAM + Restarts finds a correct solution in all 5 trials with a median time of 38 seconds, but the solutions are slightly different (the simplified solutions are listed):

```
ZipWith(lambda u1, u2: Min(4, Max(0, u2)), x1, x1)
ZipWith(lambda u1, u2: Min(4, Max(0, u1)), x1, x1)
Reverse(ZipWith(lambda u1, u2: Min(4, Max(0, u2)), x1, Reverse(x1)))
Reverse(Map(lambda u1: Min(4, Max(0, u1)), Reverse(x1)))   # found in two trials
```

Note that these are not the shortest solutions, but nevertheless all of these solutions are equivalent to the ground-truth solution. LAMBDABEAM's solutions could benefit from a postprocessing simplification step, as discussed in Section 4.3.

# E   More Details on LAMBDABEAM Architecture and Training

In our experiments, we used the following hyperparameters for the LAMBDABEAM model architecture and training procedure. Refer to Figure 2 for a diagram showing how the different modules interact.

- I/O Module: this encodes a property signature of the I/O examples using a 2-layer ReLU-MLP with hidden size and output size of 512.
- Value Module: this encodes each value's property signature using a 2-layer ReLU-MLP with hidden size of 512 and output (embedding) size of 256, with a layer-norm applied after each linear projection. We use different MLPs for lambda and non-lambda expressions.
- Search Context Summary Module: this module needs to represent the entire search state at the current stage, including the current operator to be expanded, the I/O specification, and the values explored so far. We compute the average of the set of value embeddings, concatenate it with the I/O embedding, and then apply a projection layer (denoted as $MLP_{op}$ in Figure 2, which projects back to the embedding dimension) to get a vector representation. The model parameters used in the projection layers are indexed by the operator (i.e., we use different sets of trainable parameters for different operators).
- Argument Selector Module: we use an operator-specific 3-layer LSTM with hidden size of 256. The prediction head is a 2-layer MLP with hidden size of 512.
- During training, we generate on-policy data with beam size 10, use an effective batch size of 32, and use the Adam optimizer with a constant learning rate of $5 \times 10^{-4}$.
- During evaluation, we use UniqueRandomizer with beam size 10.

# F   Analysis of Handwritten and Synthetic Tasks

Table 1 shows some differences in the distributions between our handwritten and synthetic evaluation tasks. This analysis may help contextualize the experimental results in Section 4.

For example, Figure 5 shows that the LLM solved abnormally many synthetic tasks in the 11-12 weight bucket. In fact, for synthetic tasks of weight 8 or more, every one of the LLM's "solutions" are actually false positives using some form of "if the input is ⟨hardcoded⟩ then return ⟨hardcoded⟩" logic, which is easier to implement when the output is an integer as opposed to a list. Table 1 shows that there are abnormally many synthetic tasks of weight 11-12 that have integer outputs, which helps to explain the results.

Table 1: Analysis of some differences in task distributions between the handwritten and synthetic evaluation tasks. We show the number of tasks with each weight, the number of tasks using a lambda expression in the ground truth solution, and the number of tasks where the desired output is an integer (as opposed to a list). There may also be other axes where the handwritten and synthetic tasks have different distributions.

| Weight | 100 Handwritten Tasks | | | | 100 Synthetic Tasks | | | |
|---|---|---|---|---|---|---|---|---|
| | # Tasks | With Lambda | | Int Output | # Tasks | With Lambda | | Int Output |
| 3 | 0 | – | | – | 10 | 3 | 30% | 2 20% |
| 4 | 3 | 2 | 67% | 2 67% | 10 | 2 | 20% | 6 60% |
| 5 | 7 | 0 | 0% | 3 43% | 10 | 3 | 30% | 5 50% |
| 6 | 10 | 9 | 90% | 4 40% | 10 | 7 | 70% | 3 30% |
| 7 | 9 | 6 | 67% | 7 78% | 10 | 5 | 50% | 5 50% |
| 8 | 11 | 11 | 100% | 4 36% | 10 | 8 | 80% | 2 20% |
| 9 | 16 | 16 | 100% | 3 19% | 10 | 8 | 80% | 3 30% |
| 10 | 9 | 9 | 100% | 1 11% | 10 | 8 | 80% | 3 30% |
| 11 | 11 | 8 | 73% | 5 45% | 10 | 6 | 60% | 7 70% |
| 12 | 9 | 9 | 100% | 3 33% | 10 | 3 | 30% | 7 70% |
| 13 | 5 | 5 | 100% | 2 40% | 0 | – | | – |
| 14 | 3 | 3 | 100% | 1 33% | 0 | – | | – |
| 15 | 4 | 4 | 100% | 1 25% | 0 | – | | – |
| 16 | 2 | 2 | 100% | 0 0% | 0 | – | | – |
| 17 | 0 | – | | – | 0 | – | | – |
| 18 | 0 | – | | – | 0 | – | | – |
| 19 | 1 | 1 | 100% | 0 0% | 0 | – | | – |
| Total | 100 | 85 | 85% | 36 36% | 100 | 53 | 53% | 43 43% |

