# OpenReview forum: "LambdaBeam: Neural Program Search with Higher-Order Functions and Lambdas"
_NeurIPS.cc/2023/Conference — NeurIPS 2023 poster_

### Official Review · Reviewer_JwfZ · 2023-07-05

**Soundness:** 3 good
**Presentation:** 3 good
**Contribution:** 4 excellent
**Rating:** 7
**Confidence:** 4

**Summary:**

The paper extends the CrossBeam method with synthesizing intermediate lambda
functions to solve the programming by example task. The authors introduce the
Merge operator to construct new lambda functions by choosing an operator from
the DSL, and its arguments from existing terms (variables or lambda
functions). The inputs to the Merge operator are predicted by the neural model,
so the lambda functions are built step-by-step bottom-up, similarly to the whole
synthesized program.

Lambda functions can not be executed on the input/ouput examples so they are
executed on hardcoded canonical argument tuples instead and are encoded using
property signatures computed on the results, the expected output of the program,
and the arguments of the lambda function.


**Strengths:**

As the authors claim and also as far as I know this is the first neural search
method to synthesize general-purpose (not hardcoded) lambda functions, which has
been an open problem for years. DreamCoder (cited in the Related Work) can also
synthesize lambda functions but it does so by finding common program fragments
in synthesized programs in a Bayesian framework.

**Weaknesses:**

The paper - understandably - refers to the CrossBeam paper many times. It
contains a summary of CrossBeam in lines 100-121, but reading that paper still
helped a lot to undertand this paper. Also, I think that Figure 2 from the
CrossBeam paper should be included as Section 3.3 talks about parts of that
Figure.

It would be good to include CrossBeam without lambda functions as a baseline for
the evaluations; currently it is hard to know how much of the improvement is due
to the lambda functions.

Part 3.1 could be clearer:
- the example is at the end of the section, maybe an example-first approach
  could be better
- Merge ensures that there are no free variables and also unifies the arguments.
  When reading through the section it seems from line 128 to 147 that the only
  criterion we need is that we have no free variables and that alone ensures the
  unification of the arguments.
- I'm not sure these are correct:
  - line 157 says that Merge runs the function $f$,
  - line 160 says that $a_k(i_k)$ evaluates to a concrete value, I think it
    should be an expression

line 217 says that $S$ contains variable tokens, but I think it also contains
lambda expressions.

I'm not sure how to intepret line 223: "an embedding of the weight of this value".

I couldn't find which LLM the authors used as a baseline, it should be cited.

I couldn't find the range of the inputs and outputs, it would be good to include
them.



**Questions:**

There were 16 canonical argument tuples for each combination of tuple length and
argument types. To me it seems somewhat low. Wouldn't more be helpful, or does
the slowdown of the search counteract the improvement?

Why do most of the methods perform worse on the handwritten tasks compared to
the synthetic tasks? Is it because the generation of the synthetic tasks is
similar to their algorithm?

Figure 4 (Synthetic tasks) shows significant improvement in the LLM results as
the task weight grows from 9-10 to 11-12. Is that an anomaly or is there an
explanation? It would be interesting to see what happens after task weight
11-12.



**Limitations:**

Limitations are addressed at the end of the Results, maybe they could have their
own section.

---

> ### Author Rebuttal · Authors · 2023-08-09
>
> Thank you for your review!
>
> > include CrossBeam without lambda functions as a baseline
>
> CrossBeam would not perform well because 85/100 handwritten evaluation tasks and 53/100 synthetic evaluation tasks use a lambda function in the solution. CrossBeam would not be able to solve those problems and would end up being the worst in our comparison.
>
> > Part 3.1 could be clearer
>
> We used “unify” loosely here. We mean that the arguments $a_k$ are alpha-renamed using $i_k$ so that we can choose which variables are used in which locations. For example, if we have a term $T = \lambda v_1. (v_1 + 1)$, then this alpha-renaming step is needed to distinguish between $Merge(\times, T, [v_1], T, [v_2]) = \lambda v_1, v_2. (v_1 + 1) \times (v_2 + 1)$ and $Merge(\times, T, [v_1], T, [v_1]) = \lambda v_1. (v_1 + 1) \times (v_1 + 1)$. We will replace our usage of “unify” with “alpha-rename”.
>
> Line 157: Instead of “Merge runs $f$ on the arguments”, we should say “Merge creates an expression that calls $f$ on the arguments”.
>
> Line 160: The argument $a_k(i_k)$ is indeed an expression. We mean to point out that this expression is not immediately usable as a lambda in a higher-order function and needs to be wrapped with an explicit lambda. For example, suppose we have the term $T = \lambda v_1. v_1 + 1$. We’d use this in a first-order function as follows: $Merge(\times, T, [v_1], x, []) = \lambda v_1. (v_1 + 1) \times x$. If we use a higher-order function in the exact same way, we get $Merge(map, T, [v_1], x, []) = \lambda v_1. map(v_1 + 1, x)$ which is not valid since the first argument to map must be a function. We instead need $Merge(map, T, [u_1], x, []) = map(\lambda u_1. T(u_1), x) = map(\lambda u_1. u_1 + 1, x)$, where the $\lambda u_1.$ part is explicitly added by our definition of Merge because we know that $map$ requires its first argument to be a function with arity 1.
>
> Line 217: This sentence has ambiguous parsing which we will revise. $S$ contains variable tokens for constructing lambdas, and $S$ also contains lambda expressions and non-lambda expressions.
>
> Line 223: The weight of the value is an integer, and we look up the integer in a learned embedding table to obtain $z$. This is done in the same way as in CrossBeam (Section 3, “Value module” paragraph, in the CrossBeam paper), for both lambda and non-lambda values.
>
> > I couldn't find the range of the inputs and outputs
>
> For inputs and outputs in our handwritten and synthetic tasks, all integers are in the range [-256, 255] as in DeepCoder, and lists have lengths in the range [0, 10] which we felt was reasonable for PBE users to specify.
>
> > There were 16 canonical argument tuples ... it seems somewhat low
>
> Through profiling, we found that the majority of LambdaBeam’s time is spent computing property signatures (not running the model!), especially for lambda functions since they must be run many times. For this reason, we didn’t include too many canonical argument tuples.
>
> > Why do most of the methods perform worse on the handwritten tasks compared to the synthetic tasks?
>
> There are two potential reasons. One is actually just visual: in Figure 4, the handwritten tasks have weight buckets shifted leftward compared to the synthetic tasks, because we have handwritten tasks of weight 13-19 but all synthetic tasks have weight at most 12. For tasks of weight <= 10, LambdaBeam+Restarts performs equally well on handwritten and synthetic tasks, and $\lambda^2$ even performs better on the handwritten tasks.
>
> The other reason is that the tasks are distributed differently in the handwritten vs synthetic datasets. For example:
>
> ```
> The handwritten evaluation tasks include:
>   * 20 tasks of weight  7 -  8: 17 with lambdas, 3 without
>   * 25 tasks of weight  9 - 10: 25 with lambdas, 0 without
>   * 20 tasks of weight 11 - 12: 17 with lambdas, 3 without
>
> The synthetic evaluation tasks include:
>   * 20 tasks of weight  7 -  8: 13 with lambdas,  7 without
>   * 20 tasks of weight  9 - 10: 16 with lambdas,  4 without
>   * 20 tasks of weight 11 - 12:  9 with lambdas, 11 without
> ```
>
> The handwritten tasks include more tasks that use lambdas, and there are abnormally many synthetic tasks of weight 11-12 without lambdas which appear to be easier overall. There are likely other distribution shifts such as in the shape of the solution (deep vs wide expression trees), the operations used, or the I/O types. It is thus an impressive result that LambdaBeam+Restarts performs the best on both evaluation datasets.
>
> > Figure 4 (Synthetic tasks) shows significant improvement in the LLM results as the task weight grows from 9-10 to 11-12
>
> This is an anomaly. In fact, for synthetic tasks of weight >= 8, all of the LLM’s “solutions” are false positives with some form of “if the input is <hardcoded> then return <hardcoded>” logic. Indeed, Figure 5 shows the LLM has a very high false positive rate on synthetic tasks. This solution pattern is easier to implement when the output is an integer as opposed to a list, and there are abnormally many synthetic tasks of weight 11-12 with integer outputs.
>
> ```
> The handwritten evaluation tasks include:
>   * 20 tasks of weight  7 -  8: 11 have int output,  9 have list output
>   * 25 tasks of weight  9 - 10:  4 have int output, 21 have list output
>   * 20 tasks of weight 11 - 12:  8 have int output, 12 have list output
>
> The synthetic evaluation tasks include:
>   * 20 tasks of weight  7 -  8:  7 have int output, 13 have list output
>   * 20 tasks of weight  9 - 10:  6 have int output, 14 have list output
>   * 20 tasks of weight 11 - 12: 14 have int output,  6 have list output
> ```

---

> > ### Comment · Reviewer_JwfZ · 2023-08-14
> > **Thank you for your response**
> >
> >
> > Thank you for your insightful response.
> >
> > In Part 3.1 I was not confused by the use of the word "unify", but the structure
> > of the beginning of the section: lines 124-127 talk about equivalent expressions
> > with different variable names (renaming), then from line 128 the paper is about
> > free variables. The two parts are not connected. Renaming (or unifying) is
> > talked about again only from line 148. So my problem was that free variables
> > appear suddenly out of nowhere and seemingly there is a connection but it's not
> > explained. After reading the whole section it cleared up, but I feel it could be
> > improved. I don't think the word "unify" should be changed.
> >
> > I believe that the differences between the handwritten and synthetic tasks
> > should be mentioned in the paper and their distributions should be included in
> > the Appendix.

---

> > > ### Author Response · Authors · 2023-08-21
> > >
> > > Thanks again for all of the helpful suggestions. We will revise the text of Section 3.1 and also include our discussion on handwritten vs synthetic tasks in an appendix.

---

### Official Review · Reviewer_WhTE · 2023-07-07

**Soundness:** 3 good
**Presentation:** 3 good
**Contribution:** 3 good
**Rating:** 6
**Confidence:** 4

**Summary:**

The paper presents LAMBDABEAM, a nn-based search method for program synthesis which is built upon CROSSBEAM and can handle lambda functions and higher-order functions. Specifically, to build lambda terms, LAMBDABEAM enforces that every term constructed during search has no free variables by introducing a novel operator called MERGE. Furthermore, to learn lambda expressions, LAMBDABEAM constructs a new generalization of property signatures to represent lambda expressions. The paper shows that LAMBDABEAM outperforms existing techniques in the integer list manipulation domain (a modified DeepCoder).

**Strengths:**

1. This addresses a meaningful problem, that is how to search for lambdas and higher-order functions which would enable the synthesis of arbitrary looping computations and extend the boundary of neural program synthesis.
2. The experiment result is promising.
3. The writing is clear.
4. In the current era dominated by LLMs in the field of program synthesis and code generation, this paper makes a good attempt towards small and meaningful works. I believe that this type of work and LLMs-related works will inspire and complement each other.

**Weaknesses:**

1. Placing a figure that illustrates the LAMBDABEAM Model architecture would be better. Although the design largely follows CROSSBEAM, a LAMBDABEAM figure is necessary for showing the differences and for readers unfamiliar with CROSSBEAM.
2. Experimental settings are somewhat confusing. For example, what causes the differing number of I/O examples for list output and integer output in the hand-written?  What is the name of the pre-trained LLM since the performance gap among different LLMs on program synthesis is significant.  Can the authors further explain these settings?
3. Can the authors fine-tune the LLM on the proposed DSL which might be a better comparison?

**Questions:**

1. Will CROSSBEAM be improved with restarts?
2. Considering the elapsed time with quantitive computing resources used would be better if possible.
3. How might this kind of nn-based search method be combined with LLMs?
4. A experimental comparison with Dreamcoder is needed.

**Limitations:**

The authors have discussed the limitations in the results part.

---

> ### Author Rebuttal · Authors · 2023-08-09
>
> Thank you for your review!
>
> > Weakness 1.
>
> We will include more info about CrossBeam (see global response).
>
> > what causes the differing number of I/O examples for list output and integer output in the hand-written?
>
> In general, PBE tasks can be specified with fewer examples if the examples are more constraining. It would be easy for a program to coincidentally output the correct integer using the wrong approach, but it is less likely that an entire list matches by coincidence. For this reason, we used fewer examples (3) to specify the problem when the output is a list, and more examples (5) when the output is an integer. These numbers were chosen to avoid underspecifying the problem, while also being realistic in the number of examples a program synthesis user might want to provide.
>
> > Can the authors fine-tune the LLM on the proposed DSL
>
> It would cost a large amount of compute resources to train the LLM (a very large sequence model) on our DSL. Instead, we have trained a smaller sequence model from scratch on our DSL, which is the RobustFill approach already included in our experiments.
>
> > Will CROSSBEAM be improved with restarts?
>
> We hypothesize that CrossBeam will also improve with restarts for an appropriate restart frequency (a hyperparameter), but it is unclear how much improvement will be gained. The amount of improvement could vary by the domain or dataset used, as was the case in our experiments for handwritten vs synthetic tasks.
>
> (Also, we note that any such improvement to CrossBeam would not affect our experimental conclusions, since CrossBeam as a baseline would fail to solve a good majority of our evaluation problems which require using lambda expressions.)
>
> > Considering the elapsed time with quantitive computing resources used would be better if possible.
>
> We agree it would be better, but this is very hard to do considering the different hardware used to run CPU-only approaches ($\lambda^2$ and enumeration), CPU with GPU (LambdaBeam), mainly GPU (RobustFill), and multiple accelerators in parallel (the LLM).
>
> > How might this kind of nn-based search method be combined with LLMs?
>
> This is an excellent question for future work! There are some very recent works that use LLMs to generate programs iteratively, e.g., to self-debug their predictions (https://arxiv.org/abs/2304.05128). In a similar vein, it would be very interesting to see whether LLMs can be made to perform program synthesis search guided by other sources of info such as program evaluations.
>
> > A experimental comparison with Dreamcoder is needed.
>
> DreamCoder is fundamentally an algorithm for enriching an impoverished DSL, and shows how that enrichment process can synergize with neurally-guided program search. Therefore, it does not make sense to compare *against* DreamCoder, but to experimentally consider *augmenting* DreamCoder with LambdaBeam (using our work as DreamCoder’s neurally-guided search strategy). While that would be a fascinating avenue to explore, we believe it is sufficiently involved to not be a reasonable piece of work to include within the scope of this paper. However, we will revise the paper to include this explanation of how DreamCoder and LambdaBeam could synergize in future systems.

---

> > ### Comment · Reviewer_WhTE · 2023-08-17
> >
> > Thanks for the response! Most of my concerns have been addressed.
> >
> > My remaining concern is about "the differing number of I/O examples for list output and integer output in the hand-written". I understand that lists need fewer I/Os than integers. I'm confused about the specific choice of "3" and "5". Why not choose both of them to be "5"? Is there any cherry-pick on this choice?
> >
> > Also, I'm curious about the performance of prompt + GPT-3.5/4. (And that's why I asked questions about the name of the LLM and the combination of search+LLMs)
> >
> > Overall, it is a good paper.

---

> > > ### Author Response · Authors · 2023-08-21
> > >
> > > Thank you for your helpful comments.
> > >
> > > > My remaining concern is about "the differing number of I/O examples for list output and integer output in the hand-written". I understand that lists need fewer I/Os than integers. I'm confused about the specific choice of "3" and "5". Why not choose both of them to be "5"? Is there any cherry-pick on this choice?
> > >
> > > We wanted the benchmarks to have a good balance between being realistic (using a small number of examples, because users might not want to specify many examples) and being well-specified (we need enough examples to avoid under-specifying the task). While creating the first few handwritten tasks, we found that 3 list outputs gave a good balance, and 5 integer outputs to similarly give a good balance (note it is easier to provide examples of integer outputs). We did not use 5 list outputs because we felt the extra 2 examples were unnecessary.
> > >
> > > There was **no cherry-picking** on this choice or in the benchmarks overall. These choices were set before we started using the benchmarks during development and initial research. During development, the benchmarks were only altered to resolve clear issues (e.g., mistakes in handwritten examples) or to add more benchmark tasks.

---

### Official Review · Reviewer_c5as · 2023-07-07

**Soundness:** 3 good
**Presentation:** 2 fair
**Contribution:** 3 good
**Rating:** 7
**Confidence:** 2

**Summary:**

In this work the authors introduce LambdaBeam a method crafted to explicitly handle lambda functions and higher-order functions for neurally guided program synthesis. Towards this goal the authors first introduce a method to represent lambda functions which enables variable order independent canonical representation, and eases creation of lambda functions by merging other lambda functions. Then, the authors adapt CrossBeam a pre-exiting method for program synthesis to synthesize lambda functions, while also employing property signatures to represent lambda functions. When deployed on integer list manipulation tasks, Lambda beam surpasses other competitive baselines, in terms of both speed and success rate.

**Strengths:**

### Originality
Previous works do not explicitly model lambda functions, or learn how to compose them. This is an original contribution of the paper.

### Quality & Clarity
The paper at a paragraph level is well written.

### Significance
The approach towards representation of lambda functions is a interesting, useful and novel contribution that may be useful for many future program synthesis approaches.
Furthermore, the approach is able to beat strong baselines such as lambda^2 an off-the-shelf program synthesis tool, and a 60B parameter large language model.

**Weaknesses:**

### Weaknesses

1. I think the paper does not clearly justify *why* prior works cannot model lambda functions or higher-order functions. Particularly, it states Programming by examples (PBE) demands a more systematic search strategy but its unclear why that is the case. The paper also doubles down on the belief that other methods *cannot* synthesize programs with arbitrary looping computation, but its unclear why that is the case. For example, large scale models (GPT 3.5) can indeed produce programs with higher-order functions and lambda functions. Its unclear why the paper strongly posits that other methods cannot do this.

2. The paper is not easy to understand and seems to depend on the reader being familiar with CrossBeam> On that note, LambdaBeam strongly depends on CrossBeam which is a small drawback as well (though the other contribution - representation of lambda functions is a general and useful contribution).



**Questions:**

Why did the authors use DeepCoder benchmark? Is integer manipulation benchmark the right choice when solutions many involve a lot of hierarchical reasoning. I am especially concerned since the input contains only 3-5 examples, which might not be sufficient to triangulate the true function. Furthermore, the False Positive rate might be higher simply because of the benchmark used.




**Limitations:**

The paper does not discuss limitation or potential negative societal impact. Adding information regarding both these aspects in the appendix might further improve the paper.

---

> ### Author Rebuttal · Authors · 2023-08-09
>
> Thank you for your review!
>
> > Weakness 1.
>
> We were careful with our wording, but we will revise to make it more clear. When we discuss previous methods that cannot handle lambda functions (line 23), we are referring specifically to the types of prior works listed on lines 22-23, NOT referring to LLMs. We refer to this line of work, on deep learning for PBE, as “neural synthesis search”. Previous work in neural synthesis search (as cited on lines 22-23) indeed does not generate arbitrary looping computations.
>
> Why not? We discuss a key difference in lines 32-26, and two difficulties in lines 41-49 and 50-56. In short: When explicitly searching over programs, the search space is very large. The best way that has been discovered in neural synthesis search is to evaluate partial programs, and use the result as input to the neural search policy. When building up a lambda function during search, the search method does not know yet what the inputs to the lambda will be, so we cannot directly evaluate the lambda. How might we represent lambdas efficiently (pruning the search space) and provide evaluation information to the neural search policy? These two difficulties are addressed by key contributions of our paper.
>
> As for LLMs: Yes, LLMs *can* produce programs with loops, and quite impressively, but they perform synthesis from _natural language_ which is very different from PBE where we have only input-output examples. LLMs seem to depend strongly on natural language for their success. When given input-output examples alone (i.e., the PBE setting), we find in our results that LLMs perform poorly. This is why we say “PBE demands a more systematic search strategy”.
>
> To our knowledge, LambdaBeam is the first neural synthesis search method that handles lambda functions or arbitrary looping computations. Note that $\lambda^2$ handles lambdas but is not neural, and language models also handle loops and lambdas but do not perform any clever search (other than beam search or rejection sampling), and hence have trouble with PBE.
>
> > Weakness 2.
>
> We will include more info about CrossBeam (see global response).
>
> > Why did the authors use DeepCoder benchmark?
>
> We needed a benchmark that included arbitrary lambda functions, and the DeepCoder benchmark was the closest to that since it had hardcoded lambda functions. Thus, we extended DeepCoder to allow arbitrary lambda functions.
>
> For the 100 handwritten evaluation tasks, we generally found that 3-5 examples are enough to sufficiently describe the task, and that false positive solutions are generally complicated unnatural programs that are accidentally correct on the examples. However, a higher proportion of the synthetic tasks are actually ambiguous, which can be seen from the significantly higher false positive rate on synthetic tasks compared to handwritten tasks.
>
> > The paper does not discuss limitation
>
> We discuss limitations on lines 342 - 347.

---

> > ### Comment · Reviewer_c5as · 2023-08-14
> > **Post-Rebuttal update**
> >
> >
> > I thank the authors for the detailed rebuttal. This rebuttal, along with other reviewer's notes has been useful to improve my understanding of this work.
> > > Weakness 1
> >
> > "Neural synthesis search" generally covers a much wider space than "deep learning for PBE", and is likely to be misleading for readers. I believe the paper would be improved by making the claims more specific.
> >
> > Example Excerpt from lines 29-31:
> >
> > "The fundamental question explored in this paper is whether a neural program synthesis search policy can learn to reason about lambdas and higher-order functions, which would enable the synthesis of arbitrary looping computations that were not previously possible with neural synthesis search" -> not possible with neural synthesis search techniques which relies on intermediate expression evaluation.
> >
> > Also, any auto-regressive token-wise prediction approach which does not rely on intermediate expression evaluations can model lambda functions (**not just LLMs**). i.e. transformer models which perform program synthesis via a simple next-token-prediction task (as done in PLAD [1]), **without relying on natural language**, can indeed predict lambda functions (if its trained on examples containing lambda functions).
> >
> > > Weakness 2
> >
> > I appreciate the authors response!
> >
> > > DeepCoder Benchmark
> >
> > I appreciate the authors response! Its indeed true that the handwritten benchmark has a smaller false-positive rate (which reflects well on the proposed method). I can understand the reasons for using this benchmark. Hopefully, in the future works more suitable benchmarks are employed.
> >
> > > Limitations
> >
> > I thank the authors for correcting my statement. The paper does mention the limitations of the proposed approach. I would still suggest the authors add potential negative societal impact in the appendix.
> >
> >
> > The authors have addressed my queries. Therefore, I am raising my rating to 7 Accept (on the expectation that authors will edit the draft to be more specific about the paper's contribution w.r.t. prior work such as in Lines 29-31).
> >
> > ### Reference
> >
> > [1] PLAD: Learning to Infer Shape Programs with Pseudo-Labels and Approximate Distributions, R. Kenny Jones et al., CVPR 2022.

---

> > > ### Author Response · Authors · 2023-08-21
> > >
> > > Thanks again for your helpful suggestions and discussion. We will clarify lines 29-31. Yes, autoregressive sequence models can predict lambda functions if trained on them; after all, our RobustFill comparison does exactly this, so we agree that it's important to be clear about this.

---

### Official Review · Reviewer_ErYC · 2023-07-07

**Soundness:** 3 good
**Presentation:** 3 good
**Contribution:** 3 good
**Rating:** 7
**Confidence:** 4

**Summary:**

This paper presents a method for training a neural module to guide a search-based program synthesis procedure that supports lambda functions. This is accomplished by leveraging the existing technique of property signatures, which essentially represent program constructs using a hand-designed vector of features. The authors design features for representing lambda functions, including evaluating the lambda function on a hardcoded set of inputs. This allows lambda functions to be incorporated in a prior bottom up program search technique called CrossBeam, and they introduce a new Merge operator to build lambda terms from the bottom up. The results on a modified version of the DeepCoder benchmark show their approach outperforms several strong baselines, including symbolic search and an LLM fine-tuned on Python.

**Strengths:**

Representing and synthesizing (lambda) functions is a significant step forward for neural program synthesis. Although the individual techniques are mostly prior art, I consider the combination of techniques to be novel.

The writing is generally clear, though I found the implementation details to be quite sparse in places (see Weaknesses)

**Weaknesses:**

The biggest weakness is that the approach is tested on only one synthetic dataset and also relies on hand designed features. Furthermore, the lambda functions only contain 2 variables. It remains to be seen if this approach could scale to more realistic datasets with more data types and more complex functions.

Additionally, there are almost no examples in the paper. At the very least, the appendix should include some examples of the synthesis dataset as well as programs synthesized by LambdaBeam.

Finally, many of the implementation details are not listed out fully, which impedes reproducibility (e.g., the architecture of the models) and in severe cases, the reader's ability to contextualize the results (e.g., a list of the property signatures used and the evaluation tasks).

**Questions:**

How many programs are there of weight at most 12 (i.e., that were sampled from for the training set)?

What size is the property signature? (or how many properties are there)

Can you elaborate on why you add an embedding of the weight of a value for embedding lambda expressions?

Is Merge complete for lambda functions? This should be addressed in the paper.

**Limitations:**

The authors should also address the extent to which the property signatures are tailored to the DSL, and the implications for broader applicability / scalability.

---

> ### Author Rebuttal · Authors · 2023-08-09
>
> Thank you for your review!
>
> > there are almost no examples in the paper
>
> This is a great point. Please see the global response for some example tasks and synthesized programs.
>
> > many of the implementation details are not listed out fully, which impedes reproducibility (e.g., the architecture of the models)
>
> We provide more details below, and we will add them to the paper:
>
> * I/O encoder: it encodes the property signatures of I/O examples using a 2-layer relu-MLP with hidden size and output size of 512.
> * Value Module: it encodes the property signatures of a value using a 2-layer relu-MLP with hidden size of 512 and output (embedding) size of 256, with layer-norm applied after each linear projection. We have different MLPs for non-lambda and lambda expressions, as mentioned in Line 224 in the paper.
> * Argument Selector Module: we use an operator-specific 3-layer LSTM with hidden size 256. The prediction head is a 2-layer MLP of hidden size 512.
>
> During training, we use beam size 10 to generate on-policy data, where the effective batch size is 32 and a constant learning rate of 5e-4 is used with the Adam optimizer.
>
> We will release our code and model checkpoints if accepted, to aid in reproducibility.
>
> > The biggest weakness is that the approach ... also relies on hand designed features
>
> > a list of the property signatures used
>
> > What size is the property signature?
>
> We indeed use some hand-designed features. However, our approach actually requires less manual design than it might seem, because we devised a system of combinatorially combining property functions to greatly expand the richness of the property signatures. Appendix A (in the supplementary material) describes this in full detail.
>
> In particular, in total across types in the DSL, we defined 20 “basic properties” of objects, 8 other objects “relevant” to understanding an object, and 24 properties for comparing two objects of the same type. These are all explicitly listed in Appendix A.
>
> With only these building blocks, using the compositional approach in Appendix A, we encode lambda values with property signatures of length 558, and encode non-lambda values with property signatures of length 359.
>
> > How many programs are there of weight at most 12 (i.e., that were sampled from for the training set)?
>
> When generating training data, we first randomly generate some inputs to a synthesis problem, i.e., the input variables’ values across multiple examples (line 258). Then, the cardinality of the space of training programs varies depending on the random inputs (the number of inputs and their types), and because we exclude suboptimal solutions from the training set. That is, a program is excluded from the training set if there is a different program found earlier in our enumerative search that evaluates the same way on the input variables for all examples.
>
> That said, we can still provide ballpark numbers. Consider the “map:replace” task from the global response. Then, starting from those inputs and ignoring the task’s outputs, our baseline enumeration finds the following programs (all with different behavior when run on the examples):
> * 313,842 programs with weight at most 8, in 5 minutes
> * 1,573,527 programs with weight at most 9, in 30 minutes
> * 8,390,593 programs with weight at most 10, in 170 minutes
>
> When generating training data, not all searches reach weight 12 within the 1 hour time limit (depending on the random inputs).
>
> Of course, there would be many more programs if we relax the constraint of solution optimality, and there are even more programs that are syntactically valid but fail to typecheck.
>
> > Can you elaborate on why you add an embedding of the weight of a value for embedding lambda expressions?
>
> This is carried over from CrossBeam. Every value in CrossBeam and LambdaBeam (including lambda and non-lambda expressions) has an embedding of the value’s weight added to the embedding of the value. This helps the model understand the “cost” associated with using this value which may influence the model’s decisions, e.g., to not use values that have too high of weight. This may help the model avoid getting stuck exploring a “rabbit hole” of larger and larger expressions that ultimately do not lead to progress.
>
> > Is Merge complete for lambda functions?
>
> Yes, Merge is complete, in the sense that we can use it to generate any function of the inputs in our DSL. Let $x_1 \dots x_n$ represent the inputs to the programming-by-example (PBE) task. All solutions to the PBE task are a function $\lambda x_1 \dots x_n. t$, where $t$ has no unbound variables other than $x_1 \dots x_n$. We can show that Merge generates the set of all terms $t$ in our DSL that have no unbound variables other than $x_1 \dots x_n$.
>
> Proof sketch: Use structural induction. The recursive case is where the expression has the form $t = \lambda v_1 \dots v_n. f(a_1 \ldots a_K)$. For each of the $a_k$, we define a new term $b_k$ which binds all of the free variables in $a_k$. Inductively, $b_k$ can be generated using Merge. We then set variable tuples $i_k$ appropriately such that $Merge(f, b_1, i_1, b_2, i_2, \dots)$ produces $t$. Intuitively, the $i_k$ can be seen as undoing any alpha-renaming when going from $a_k$ to $b_k$.
>
> We will clarify this in the paper, and add a formal proof in an appendix.

---

> > ### Comment · Reviewer_ErYC · 2023-08-12
> >
> > Thanks for the response! This has addressed substantially all my concerns and I will increase my score to a 7.

---

### Author Rebuttal · Authors · 2023-08-09

We appreciate all of the insightful reviews! We will revise our paper to incorporate our clarifications and new information wherever appropriate.

This global response includes information helpful for multiple reviewers, and we also respond to each reviewer individually.

**The paper could use more background on CrossBeam**
(reviewers c5as, WhTE, JwfZ)

If accepted, we will use the extra page to include more background on CrossBeam, and LambdaBeam’s relationship to CrossBeam (e.g., with a figure), to help alleviate this weakness.

**Which pre-trained LLM is used?**
(reviewers WhTE, JwfZ)

We omitted the name and citation for double-blind purposes. We will certainly include more details and citation if accepted.

**Minor correction**

Line 260 says each synthetic training task has between 3 and 5 I/O examples. Actually, synthetic tasks have between **2** and 5 I/O examples.

**The paper could use examples of evaluation tasks and synthesized programs**
(reviewer ErYC, but we think this context will be helpful to everyone)

We will add some evaluation tasks and synthesized programs to an appendix. We cannot upload new revisions during the review period, so we provide some interesting examples here. The task names are only for convenience and are not used in our comparisons.

**“map:replace”**
This handwritten task has 3 inputs (`x`, `f`, and `r`), 3 examples demonstrating the task (“in `x`, find instances of `f` and replace them with `r`”), and a handwritten ground-truth solution using a relatively complicated lambda function:
```
inputs_dict={
    'x': [[7, 2, 4, 6, 4, 2, 5],
          [-6, -3, 4, 3, -5, -3, 2, 1, 5],
          [18, 48, 27, 26, 27, 27, 28, 17, 27, 33]],
    'f': [4, -3, 27],
    'r': [-1, 7, 99],
}
outputs=[[7, 2, -1, 6, -1, 2, 5],
         [-6, 7, 4, 3, -5, 7, 2, 1, 5],
         [18, 48, 99, 26, 99, 99, 28, 17, 99, 33]]
solution='Map(lambda u1: If(Equal(u1, f), r, u1), x)'
```
In `inputs_dict`, each of the entries for `'x'`, `'f'`, and `'r'` is a list of length 3, which contains the input for each of the 3 examples.

LambdaBeam+Restarts finds the same solution (weight 10) in each of the 5 trials: `Map(lambda u1: (lambda v1: If((lambda v1: Equal(f, v1))(v1), r, v1))(u1), x)`, taking a median time of 202 seconds.

The solution looks complicated due to the Merge operation causing lots of variable renames (i.e., $a_k(i_k)$ in the Merge definition). We have implemented an algorithm to simplify the solution by statically resolving these renames. In this case, the solution simplifies to `Map(lambda u1: If(Equal(f, u1), r, u1), x)` which is essentially identical to the ground-truth solution.

**“multi:multiply_odds”**
This handwritten task has 1 input and uses multiple higher-order functions to compute a running product of only the odd elements:
```
inputs_dict={
    'x': [[3, 5, 8, 2, 1],
          [5, 2, 1, 3, 3, 1, 4],
          [3, -4, -1, 8, 2, 0, -3, 0, 9, -1]],
}
outputs=[[3, 15, 15],
         [5, 5, 15, 45, 45],
         [3, -3, 9, 81, -81]]
solution='Scanl1(lambda u1, u2: Multiply(u1, u2), Filter(lambda u1: IsOdd(u1), x))'
```

In each of the 5 trials, LambdaBeam+Restarts finds the same solution (weight 11) that simplifies to the ground-truth solution, taking a median time of 75 seconds.

**“synthetic:weight_9_function_7”**
This synthetic task clips every element to the range [0, 4]:
```
inputs_dict={
    'x1': [[-9, -2, -10, -6, 0, -10, -6, 3, 1],
           [-1, -5, 8, 5]]
}
outputs=[[0, 0, 0, 0, 0, 0, 0, 3, 1],
         [0, 0, 4, 4]]
solution='Map(lambda u1: Min(4, Max(0, u1)), x1)'
```

LambdaBeam+Restarts finds a correct solution in all 5 trials with a median time of 38 seconds, but the solutions are slightly different (the simplified solutions are listed):
* `ZipWith(lambda u1, u2: Min(4, Max(0, u2)), x1, x1)`
* `ZipWith(lambda u1, u2: Min(4, Max(0, u1)), x1, x1)`
* `Reverse(ZipWith(lambda u1, u2: Min(4, Max(0, u2)), x1, Reverse(x1)))`
* `Reverse(Map(lambda u1: Min(4, Max(0, u1)), Reverse(x1)))` (this solution is found in two trials)

Note that these are not the shortest solutions, but nevertheless all of these solutions are equivalent to the ground-truth solution. LambdaBeam solutions could benefit from a postprocessing simplification step (line 339) but this is orthogonal to our contributions.

---

### Decision · Program_Chairs · 2023-09-21

**Decision:**

Accept (poster)

**Comment:**

This paper presents a method for doing synthesis in the integer list manipulation domain (DeepCoder) with higher-order functions.  One new technique needed to do this deals with handling equivalent lambda representations for efficiency of pruning, which relies on a new MERGE operation. The paper also uses property signatures to analyze how lambdas are used and represent them appropriately in the neural model.

The reviewers found the paper to be tackling an interesting and novel topic, of how to model lambdas in program synthesis. They praised the execution of the method and the experimental results. The biggest weaknesses of the paper that were raised are (1) its dependence on CrossBeam, and its nature as an extension of that method, which primarily impacts the writing and the ability of this paper to stand alone in its current form. (2) The results on a single synthetic dataset, which makes the technique feel a bit unproven in broader or more realistic settings.